

# Combined use of volume radar observations and high-resolution numerical weather predictions to estimate precipitation at the ground: methodology and proof of concept

Tony Le Bastard[1], Olivier Caumont[2], Nicolas Gaussiat[1], and Fatima Karbou[3]

[1]DSO-CMR, Météo-France, Toulouse, France
[2]CNRM, Université de Toulouse, Météo-France, CNRS, Toulouse, France
[3]CNRM-CEN, UMR 3589, Météo-France, CNRS, St. Martin d'Hères, France

**Correspondence:** Tony Le Bastard (tony.lebastard@meteo.fr)

**Abstract.** The extrapolation of the precipitation to the ground from radar reflectivities measured at the beam altitude is one of the most delicate phases of radar data processing for producing Quantitative Precipitation Estimations (QPEs) and remains a major scientific issue. In many operational meteorological services such as Météo-France, a Vertical Profile of Reflectivity (VPR) correction is uniformly applied over a large part or the entire radar domain. This method is computationally efficient and

the overall bias induced by the bright band is most of the time well corrected. However, this way of proceeding is questionable in situations with high spatial and vertical variability of precipitation (during the passage of a cold front or in a complex terrain, for example).

This study initiates from two statements: first, radars provide information on precipitation with a high spatio-temporal resolution but still require VPR corrections to extrapolate rain rates at the ground level. Second, the horizontal resolution of some

Numerical Weather Prediction (NWP) models is now comparable with the radar one and their dynamical core and microphysics schemes allow to produce realistic simulations of VPRs.

The present paper proposes a new approach to assess surface rainfall from radar reflectivity aloft by exploiting simulated VPRs and rainfall forecasts from the high resolution NWP model AROME-NWC. To our knowledge, this is the first time that simulated precipitation profiles from a NWP model are used to derive radar QPEs.

The implementation of the new method on two stratiform situations provided significant improvements on the hourly and 6-h accumulations compared to the operational QPEs, showing the relevance of this new approach.

## 1   Introduction

Precise quantitative precipitation estimates (QPEs) are critical for many applications including nowcasting, hydrology and flood forecasts. For instance, high resolution QPEs are necessary to correctly predict the occurrence and intensity of heavy

rainfall events and flash floods. Operational rain gauges can provide accurate measurements of the rainfall, but the operational networks are generally too sparse to capture the high spatial and temporal variability of precipitation, especially over complex terrain.



Weather radars provide spatially continuous precipitation estimates at very high spatial and temporal resolutions, thus filling the gaps in in-situ observation networks. However, the radar does not provide direct estimates of precipitation but rather an indirect measure which is the backscattered power of hydrometeors in the atmosphere. Hydrometeors may be liquid or solid or a mixture of two states and can precipitate to the ground in the form of rain or snow depending on several factors. The

estimation of precipitation by radar is therefore subject to several sources of error that need to be understood and evaluated (Zawadzki, 1984; Joss and Lee, 1995; Dinku et al., 2002; Villarini and Krajewski, 2010).

These errors are of three kinds: (i) measurement errors of the radar moments ($Z$, $Z_{dr}$, $\phi_{dp}$), (ii) conversion errors in the quantitative precipitation estimation of the precipitation from the radar moments and (iii) extrapolation errors in the determination of the precipitation falling at the ground from the estimations obtained at beam heights. In recent years, significant progresses

have been made to reduce the first two types of errors by better controlling the quality of the polarimetric parameters (calibration of $Z$ and $Z_{dr}$, adaptive smoothing of $\phi_{dp}$, attenuation correction) (Bringi et al., 2001; Gourley et al., 2009; Yu et al., 2018) and by combining the polarimetric moments to estimate the precipitation with minimal uncertainty (Ryzhkov et al., 2005; Tabary et al., 2011; Figueras i Ventura et al., 2012). However, the final step of estimating the precipitation at the ground remains a major challenge in particular in mountainous regions where lower beams can be partially or totally blocked (Creutin

et al., 1997; Smith, 1998).

Many National Weather Services have implemented a Vertical Profile of Reflectivity (VPR) correction that either uses a climatological profile obtained from a large number of radar observations over a period of time (Andrieu and Creutin, 1995; Vignal et al., 1999; Borga et al., 2000; Seo et al., 2000; Germann and Joss, 2002; Kirstetter et al., 2010) or uses an idealized profile adjusted in real time (Kitchen et al., 1994; Tabary, 2007) with a pixel-wise approach or not. In both correction methods, the

VPR can only be retrieved to represent the volume of atmosphere sampled by the radar and thus cannot inform on the vertical structure of the precipitation in shielded areas. As a consequence, any evaporation or enhancement of precipitation occurring below the lowest usable beam is missed and can lead in turn to an over- or an under-estimation of the precipitation at the surface (Gori and Joss, 1980; Hu and Srivastava, 1995; Li and Srivastava, 2001).

Bauer et al. (2015) suggested that high resolution NWP model could bring useful information about the vertical structure of the

precipitation which could help to address the shortcomings of radar measurements. Thus, they tried to directly retrieve QPEs from NWP model simulations (WRF) including radar data assimilation. But the developed method could not compete with observation-based QPE methods in part because of the weaknesses of the assimilation system and the model spin-up. So far the use of simulated data from NWP for producing radar-based QPE has generally been limited to the freezing level height for determining the top of the melting layer (Kitchen et al., 1994; Tabary, 2007).

More recently, some attempts to closely exploit NWP outputs have been done to assess the VPR shape in the lower levels. For instance, Georgiou et al. (2010) have proposed a parametrized orographic enhancement term to represent the seeder-feeder mechanism occurring over hills and small mountains (Purdy et al., 2005) that can be added to VPR correction. More recently, Martinaitis et al. (2018) have proposed a real-time evaporation correction scheme using environmental temperature and humidity from a NWP model.

Previous studies have shown that synthetic, yet realistic radar observations can be obtained by applying a radar simulator to





model outputs (Caumont et al., 2006; Jung et al., 2008; Pfeifer et al., 2008; Ryzhkov et al., 2011; Wolfensberger and Berne, 2018). The VPR estimation and correction is a major step in radar data processing that determines the quality of radar precipitation that would be observed at the ground. In this paper we propose a new method allowing the estimation of VPR profiles in good coherence with meteorological and radar observation conditions. We rely on predicted profiles of precipitation of the

French high-resolution NWP model AROME-NWC (Auger et al., 2015) which are directly used together with the observed volume radar polarimetric parameters to determine the most realistic VPR over a particular location and produce a better estimate of the precipitation at the ground. The operational radar data processing for producing radar QPEs and the radar simulations are presented in the first section. Section 2 is dedicated to the presentation of the new QPE method. In section 3, the benefits of the new method are illustrated and quantified on two particular case studies.

## 10  2   Data and tools

### 2.1   Operational radar QPE

The French metropolitan radar network has a total of 31 Doppler radars (5 S-band, 20 C-Band, and 6 X-band). All except 2 are polarimetric and configured to transmit 2-$\mu$s pulses in triple pulse repetition time mode (Tabary et al., 2006). The radar acquisition software samples the reflectivity and polarimetric parameters at the resolution of 240 m $\times$ 0.5° (polar grid) up to a

maximum range of 255 km. Radar volumes are produced every 5 min and are made of 3 to 5 low elevation scans (between 0 15 and 3°) repeated every 5 min and 2 to 3 high elevation scans (between 3 and 15°) repeated every 15 min.

The centralised processing platform located in Toulouse ingests the raw elevations scans received from the radars and applies corrections for partial beam blockages, gas attenuation and radome attenuation (at X band only) to the noise thresholded reflectivity data. The processing chain uses polarimetric variables (Ventura and Tabary, 2013) to identify non-meteorological

echoes and also corrects for the attenuation of the precipitation. Finally, the polar cells are projected on a regular Cartesian 20 grid of 1 $\times$ 1 km$^2$. All the scans are synchronised to the end of the volume scan time window using a 32 $\times$ 32 km$^2$ advection field. This latter is deduced from the two latest composite reflectivity fields through the search of the advection vectors that minimises the differences between the current composite and the advected previous one.

Once the scan processing is complete, a VPR correction is applied to each elevation scan to produce an estimate of the

reflectivity at the surface. The operational VPR correction (Tabary, 2007) uses an idealised VPR represented by four parameters (Fig. 1): the freezing level height (FLH), the decreasing rate (DR) of reflectivity above, the bright band thickness (BBT) and its amplitude (BBP). A new idealised VPR profile is inferred every radar cycle (5 min) using the ratios of the hourly rain rate accumulations gathered at different elevation angles and distances from the radar. By varying the four parameters, 300 ratio profiles of rain rate are built and compared to the observed rain rate accumulation ratios. The set of parameters that bring the

simulated ratios closest to those observed is used to define the VPR over the whole domain. The first guess of FLH is from the operational NWP model ARPEGE. Both FLH and BBT estimates can be further refined by using the cross-correlation coefficient $\rho_{hv}$ (Tabary et al., 2006). This method is computationally efficient and the overall bias induced by the bright band is most of the time well corrected. However, by defining a unique VPR, the spatial variability of the precipitation is ignored,





leading to significant local biases. For instance, when a cold front crosses the radar domain, differences in bright band heights exceeding 1,000 m can be observed between the front and the rear of the precipitating area. A unique VPR, and consequently a unique freezing level height, will necessarily generate over- and/or underestimations of the bright band altitude in some parts of the domain, leading to significant over- and/or under-correction of the reflectivity. Also, by hypothesising a constant

rain rate between the bottom of the bright band and the surface, ordinary processes such as evaporation or enhancement of the precipitation are not well captured. Thus, to compensate the limitations of the VPR correction, hourly rain gauge and radar data from the past hours (up to 40 h) are used in the operational processing to adjust the final 5-min QPE product. The adjustment factor is defined as the ratio between the total rain gauge accumulation and the corresponding radar accumulation (with greater weight given to the most recent hourly accumulation). Only rain gauge accumulation within 100 km from the radar and larger

than 0.6 mm are used in the calculation. The QPE processing used for comparison in this study (referred to as "Panthere QPE") is identical to the operational QPE processing except for the adjustment that for the sake of simplification is calculated using the accumulation ratios from the current hour only.

## 2.2   AROME-NWC and the radar forward operator

The novelty of the method presented in this paper is to take the simulated VPRs produced by the French operational model

AROME-NWC (Auger et al., 2015) as input. This new high-resolution model, especially designed for nowcasting, is based on the AROME mesoscale model (Seity et al., 2011; Brousseau et al., 2016) which provides the lateral boundary conditions and the first-guess file. AROME-NWC is run every hour and produces short-range forecast outputs up to 6 h on a domain covering France and adjacent areas. The vertical grid (90 levels) is stretched from 10 to around 30,000 m above the ground and the horizontal resolution is $1.3 \times 1.3$ km$^2$. AROME is a non-hydrostatic and convection-permitting model using a one-moment

bulk microphysical scheme called ICE3 which predicts the contents of six water species: vapour, cloud water, rainwater, graupel, snow aggregates and pristine ice. The temperature and hydrometeor contents from AROME-NWC are taken as input for a polarimetric radar forward operator (Caumont et al., 2006; Augros et al., 2016) that simulates the horizontal reflectivity $Z_{hh}$ (in dBZ unit) as well as the following polarimetric variables: the differential reflectivity ($Z_{dr}$), the differential propagation phase shift ($\phi_{dp}$), the specific differential phase ($K_{dp}$), the copolar correlation coefficient ($\rho_{hv}$), the specific and differential

attenuations ($A_{hh}$ and $A_{dp}$) and the back-scattering differential phase ($\delta_{hv}$). The horizontal regular grid of the model ($1.3 \times 1.3$ km$^2$) is preserved, but within the vertical columns, those variables are projected onto the radar beam geometry by the forward model that, for a given elevation, takes into account the bending and broadening of the beam. Rain, graupel and snow particles are modelled as oblate spheroids, while pristine ice particles are modelled as spheres because of their random orientation. Their back- and forward-scattering coefficients are computed following the transition matrix (T-matrix) method (Mishchenko et al.,

1996). For better efficiency, the operator uses T-matrix lookup tables computed in advance for each hydrometeor type and radar wavelength (X, S and C-band). In order to only retrieve observable values of reflectivity, all the reflectivities that fall below the radar minimum detectable reflectivity ($Z_{min\_detect}$) are set to "no detection". In the current radar processing chain, $Z_{min\_detect}$





is defined as:

$$Z_{min\_detect}(r) = Z_{noise\_100km} + 20.\log(\frac{r}{100}) + \delta,$$
(1)

where $Z_{noise\_100km}$ is the noise equivalent reflectivity at 100 km range, $r$ the distance from the radar in kilometres and $\delta$ the threshold used to reject noisy pixels. Both $Z_{noise\_100km}$ and $\delta$ are radar dependent. With the same forward model and a similar research model (Meso-NH) at a resolution of $2.5 \times 2.5$ km$^2$ implemented with the one-moment bulk microphysical scheme ICE3 (identical to AROME), Augros et al. (2016) have made statistical comparisons between the observed and the simulated radar variables on two Mediterranean convective events. They showed that the polarimetric forward operator produces reflectivities in overall general good agreement with observed ones. However, some discrepancies were found, especially in lower atmosphere where the simulated reflectivities tend to be underestimated which may be due to the coarser horizontal resolution of the simulation compared to the radar observations.

## 3   New radar and model combined QPE method

To address the limitation of the current VPR correction a new methodology is proposed. This method consists in finding the simulated apparent VPR from the AROME-NWC forecast (hereafter called pseudo-observed VPR or POVPR) most resembling to the observed apparent one for every radar pixel, and then to retrieve the corresponding QPE (hereafter called POVPR QPE).

### 3.1   VPR estimation

The VPR estimation is applied separately for each $1 \times 1$ km$^2$ pixel pi of the radar domain. The method developed in this study to retrieve VPRs is based on the Bayesian approach used by Kummerow et al. (1996, 2001) in the Goddard profiling algorithm (GPROF). The latter was also used by Caumont et al. (2010), Augros et al. (2018) and Borderies et al. (2018a, b) for the validation and assimilation of radar reflectivity and dual-polarisation observations in the French high-resolution model AROME. In the same way, we use here a large database made of simulated profiles $VPR_{mod}$ in the vicinity of the considered radar pixel $p_i$ to find the most probable VPR ($POVPR(p_i)$) given the observed apparent $VPR_{rad}$. Thus, $POVPR(p_i)$ is defined as a linear combination of the $VPR_{mod}$ weighted by a factor $P$ depending on the distance $d$ in terms of reflectivity between the apparent simulated VPRs ($VPR_{mod,app}$), i.e. the projection of $VPR_{mod}$ on the available radar elevations, and $VPR_{rad}$:

$$POVPR(p_i) = \frac{1}{\sum_j P(j)} \sum_j P(j).VPR_{mod}(j),$$
(2)

where

$$P(j) = \exp\left(-\frac{1}{2}d[VPR_{mod,app}(j)]\right),$$
(3)

$$d[[VPR_{mod,app}(j)] = \frac{1}{n_{elev}} \sum_{n_{elev}} \left[\frac{Z_{hh\_mod,elev}(j) - Z_{hh\_rad,elev}}{\sigma(h_{elev})}\right]^2, \text{ and}$$
(4)

$$\frac{1}{\sigma(h_{elev})^2} = \left[ \frac{2}{\frac{h_{elev}}{Alt_{max}} + 1} - 1 \right]^2 . \tag{5}$$

$j$ is the index of the profiles $VPR_{mod}$ in the vicinity of the profile $VPR_{rad}$, $n_{elev}$ is the number of elevations where the reflectivity is valid, $Z_{hh\_mod,elev}$ and $Z_{hh\_rad,elev}$ are the simulated and observed reflectivities respectively, and $\sigma(h_{elev})$ is

a weighting function depending on the height of the elevation $h_{elev}$ normalized by the maximum altitude of the radar data $Alt_{max}$ (set to 12,000 m ASL in this study). This formulation of $\sigma$ permits to give more weight to the lowest elevations. $P$ is equal to 1 for a perfect simulated apparent VPR ($VPR_{mod,app}$ = $VPR_{rad}$) and tends towards 0 as the difference between VPRs increases.

This procedure is repeated for each pixel $p_i$ of the area covered by the radar. Some settings are imposed to, on one hand, limit

the time computing, and on the other hand, to help the algorithm find the most appropriate VPR. First and foremost, simulated reflectivities below $Z_{min\_detect}(pi)$ (see Sect. 1.1.1) are considered as non-precipitating. Then, the distance for exploring the simulated VPRs around $p_i$ is set to a maximum of 100 km as proposed by Augros et al. (2018). At the full resolution of AROME-NWC (1.3 km), it represents more than 18,000 simulated VPRs to analyse. We chose to keep the spatial extent of the data set to take into account space-shifting of the simulated precipitation with respect to the observations. Thus, to reduce

the data set and consequently the computing time, only one point out of four of the simulation in each horizontal direction has been used, dropping the number of VPRs to analyse to about 1,200. Moreover, for every observed and simulated vertical profile of reflectivity that contains a non-precipitating layer surrounded by precipitating layers, only the lower precipitating part is kept. By doing so, we make the hypothesis that the lower precipitating layer is unrelated to the upper one. We also forced the selected VPRs to be in the same air mass as that in $p_i$ through a condition on the freezing level. The isotherm 0 °C at $p_i$

location is estimated by the one of the co-located point of the hourly AROME analysis. VPRs that have a freezing level height 300 m higher or lower than that in $p_i$ are excluded from the data set. Additionally, simulated VPRs where the ground level is higher by more than 300 m above the ground level of $p_i$ are also eliminated from the data set. Indeed, VPRs from higher terrain are, by nature, not defined at the ground altitude of $p_i$ and consequently cannot provide relevant information about the vertical structure of the precipitation at this altitude. On the contrary, keeping VPRs from lower terrain allows us to potentially extend

the simulated VPR data set for mountainous pixels. Finally, in order to give more importance to the nearby VPRs, only the 100 closest remaining VPRs are kept. In the hypothesis in which no VPR would be excluded from the initial data set (1,200 VPRs) during the previous filtering steps described above, this last one step would keep VPRs distant of less than 30 km from $p_i$.

### 3.2  Model bias correction

To maximise the chances of finding the simulated apparent VPRs that best fit the observations, a model bias correction is used

to bring a maximum number of simulated observations as close as possible to the observed ones. Simulated observations can be biased either because the model itself is biased (approximation in the model physics, representativeness errors) or because the radar forward operator is biased or both. For simplicity, all the model biases are corrected by applying a quantile mapping correction (QM), a method commonly used in climatic simulations (Lafon et al., 2013). Thus, this correction is applied every





hour and is used to match the distributions ($D_{mod}$) of the simulated reflectivities produced by the model ($Z_{hh\_mod}$) with the observed distributions ($D_{rad}$) computed by aggregating all 5-min radar reflectivity scans during the hour centred around the model time. The chosen 1h time window ensures that, the range of values of each 5-min observed reflectivity data set processed by the POVPR algorithm, is covered by the closest in time simulated reflectivity data set used for the VPR estimation. Some
tests (not presented here) have shown that a longer temporal window gives poorer final QPE results.

To take into account the positioning errors of the simulated precipitating columns compared to the observations, a first step consists in adjusting the distances of these columns from the radar also by quantile mapping before the projection on the radar beam geometry. Indeed, let us suppose the observation is made of a unique precipitating column $C_o$ which is perfectly represented by the simulated column $C_s$ ($C_o = C_s$) but further from the radar. Due to the different beam altitudes and widths at
the column locations, the projection of $C_s$ on the radar beam geometry will be most probably different from the projection of $C_o$ even though the unprojected columns are identical. This statement can be generalised to larger data sets. Thus, the preliminary quantile mapping correction on distances ensures that the proportions of precipitating columns at a particular distance from the radar are the same in $D_{rad}$ and $D_{mod}$ data set, which mitigates the effects of the model positioning errors. Furthermore, to have comparable samples, simulated reflectivities below theoretical noise level are eliminated and each elevation of the model
data set is repeated as often as it is present in the radar data set. Finally, the transfer function $T_{Mod \rightarrow Rad}$ from $D_{mod}$ to $D_{rad}$ is evaluated and applied to simulated reflectivities to produce the corrected reflectivity ($Z_{hh\_mod\_cor}$) data set:

$$Z_{hh\_mod\_cor} = T_{Mod \rightarrow Rad}(Z_{hh\_mod}). \tag{6}$$

In a different way, we can express $Z_{hh\_mod\_cor}$ as the sum of $Z_{hh\_mod}$ and a corresponding reflectivity correction ($C_z$):

$$Z_{hh\_mod\_cor} = Z_{hh\_mod} + C_z(Z_{hh\_mod}), \tag{7}$$

where

$$C_z(Z_{hh\_mod}) = T_{Mod \rightarrow Rad}(Z_{hh\_mod}) - Z_{hh\_mod}. \tag{8}$$

Note that the projection of simulated reflectivities is adapted to each radar pixel geometry during the VPR estimation (see next part). As a consequence, simulated reflectivities used for the VPR search can outreach the ones used for the build of $T_{Mod \rightarrow Rad}$. That is why, for reflectivities exceeding the maximum reflectivity of the simulation data set ($Z_{hh\_modmax}$), $C_z$ is
set to the correction value of $Z_{hh\_modmax}$:

$$C_z(Z_{hh\_mod} > Z_{hh\_modmax}) = C_z(Z_{hh\_modmax}). \tag{9}$$

An example of the application of the quantile mapping correction is provided in Figure 2.

For purpose of this study, the 2h lead time forecasts from AROME-NWC are used. On the one hand, this short forecast range offers the advantages to be beyond the spin-up period of the model. On the other hand, it is close enough to the analysis
time that the initial $D_{mod}$ data set can be brought as close as possible to the $D_{rad}$ data set by quantile mapping, and most representative VPR profiles can be found in the VPR estimation process. However, in situations where the simulations are





### 3.3 QPE calculation

Once the closest simulated VPRs and their corresponding weights $P$ are found for the pixel $p_i$, the rain rate at the pixel $p_i$ ($RR_{rad}(p_i)$) is estimated by the $P$-weighted linear combination of the rain rates associated to each simulated VPR ($RR_{mod}(j)$) and estimated at the same altitude as the ground altitude of $p_i$:

$$RR_{rad}(p_i) = \frac{1}{\sum_j P(j)} \sum_j P(j).RR_{mod}(j). \tag{10}$$

The entire procedure for generating $RR_{rad}(p_i)$ from radar and NWP model reflectivities is summarised in Figure 3. The 5-min precipitation accumulation $ACC(p_i)$ is simply deduced by integrating $RR_{rad}(p_i)$ over time by assuming that the rain rate is constant during this period:

$$ACC(p_i) = RR_{rad}(p_i).5min. \tag{11}$$

Operationally, a spatiotemporal interpolation of the 5-min rain rates is made at a 1-min time step to take the displacement of precipitation into account. For simplification, this has not be done here. But the stratiform nature of precipitation of the cases studied further (see Sect. 3) ensures that the error made by doing so is negligible.

Finally, similarly to the Panthere QPE calculation, we apply a simplified adjustment factor on the final radar QPEs (see Sect. 1.1.3). In a timely manner, it simultaneously reduces the impact of biases from radar measurements and simulated rain rates.

## 4 Results

Two stratiform case studies (30[th] April 2018 and 3[rd] March 2017) affecting plains areas (see Fig. 4) were chosen to demonstrate the potential benefits of the method. In these particular situations, the variability of temperature and the precipitation fields was large and the NWP model was able to produce simulations in relatively good agreement with the observations in terms of timing, localisation and intensity. The evaluation of the new QPE method with respect to the older one is first described. Then, the results are presented for both situations studied.

### 4.1 Evaluation process

To evaluate the performances of the POVPR and the operational methods, hourly and 6-h rain gauge accumulations (resolution of 0.2 mm) are compared with the co-located retrieved radar accumulations. For clarity, accumulations are removed from the data set if (i) the rain gauge has a class equal to 5 according to the WMO classification, or (ii) the radar was not able to detect any signal (signal weaker than noise level or radar beam above the precipitation), or (iii) the simulated freezing level height above the ground level is lower than 300 m (enhanced risk of snow which cannot be correctly measured by the non-heated rain



gauges from the operational network). Once the hourly and 6-h accumulations data set are built, the root mean square error (RMSE), the mean bias and the Pearson correlation coefficient (r) are computed for both radar data sets (Panthere and POVPR) by considering the rain-gauge data set as a reference. We also compute the differences between the mean hourly RMSEs from the POVPR QPEs and the mean hourly RMSEs from the Panthere QPEs according to the distance from the radar.

## 5  4.2  Back-bent occlusion of the 30$^{th}$ April 2018

On April 30$^{th}$ a quasi-stationary low concerned northern France. Its warm and cold fronts affected most part of the country and its warm sector occluded on the northernmost areas. The latter wrapped around the low, forming a so-called back-bent occlusion, which brought cold temperatures and continuous precipitation, especially in Normandy where some snow was locally witnessed. As an illustration of these features, Figure 5 displays the reflectivity and the freezing level field predicted by
AROME-NWC (4 UTC run) at 6 UTC. The heaviest rainfalls occurred between 3 and 9 UTC with accumulations up to 25 mm measured by the rain gauges.

During this event, the correctness of the accumulations computed operationally (Panthere) compared to those measured by the rain gauges was really poor (see Fig. 6a), with a strong overestimation in a corridor at a distance between roughly 45 and 85 km from the radar and a significant underestimation beyond. A deeper analysis of the lower radar scans (not shown here)
reveals all the typical features of a bright band in the area of overestimation: enhanced reflectivity, differential reflectivity ($Z_{dr}$) and specific differential phase ($K_{dp}$), as well as a low cross-correlation coefficient ($\rho_{hv}$). At 6 UTC, from the lower radar scan (0.4° elevation), the bright band top can be estimated to be at an altitude of approximately 1,000 m ASL in that location. This is consistent with the freezing level altitude predicted by AROME-NWC (see Fig. 5). However, the operational evaluation of the bright band top altitude made from the $\rho_{hv}$ radar fields prior to the determination of the VPR (see Sect. 2.2) led to a
freezing level altitude of 2,000 m ASL. Further investigations show that this overestimation is in particular due to the high spatial variability of the freezing level height combined with the radar beam geometry. The bright band in the western cold air mass is too low to be sampled by most of the radar beams. Consequently, the bright band top altitude retrieved by the radar scans is more representative of the warm air mass close to the radar location than the cold air mass further west where the QPE values are very biased.

The overestimation of the freezing level altitude has different impacts on the VPR correction (see Fig. 1) depending on the distance from the radar. Close to the radar, where the beam intercepts the bright band, reflectivities are considered to be in the rain and therefore are not corrected. It finally induces a strong overestimation of the ground rainfall accumulations. Further from the radar, where the beam is above the freezing level, three configurations lead to an underestimation of the precipitation at the ground level: (i) reflectivities are still considered to be in the rain part and are not enhanced as they should be, or (ii)
reflectivities are incorrectly flagged as lying in the bright band and are consequently wrongly reduced by the VPR correction, or (iii) reflectivities are rightly considered as snowy but are insufficiently corrected because of the underestimation of the thickness between the radar beam and the freezing level altitudes.

The POVPR method is much more reliable in this situation (see Fig. 6b-d) thanks in particular to the constraint imposed on the freezing level altitude during the research of the most appropriate simulated VPR (see Sect. 1.1). The estimations, which





are biased due to the overestimation of the freezing level altitude described above, seem to be at least partially corrected as illustrated by the difference between Panthere and POVPR accumulations (see Fig. 6c). The computed scores clearly show a significant improvement compared to the Panthere QPE. The RMSEs of the hourly and 6-h accumulations are reduced by 43 and 47% respectively, the mean biases are mitigated and approach zero, and the Pearson correlation coefficients grow from

0.60 and 0.67 respectively to 0.81 and 0.88. Those improvements are illustrated by the scatter plots of these accumulations (see Fig. 7). We can also notice that these performances are observed all along the radar range (see Fig. 8a), and especially at the ranges where the lower beam intercepts the bright band (45 to 85 km). More generally, 65% of mean hourly RMSEs at rain-gauge locations are reduced compared to the Panthere QPE ones. But these different performances cannot be fully explained by the condition on the freezing level. Indeed, the comparison between the median simulated VPR on the radar

domain relatively to the freezing level and the operational one used for the reflectivity correction shows many differences (see Fig. 9a): (i) a much more important variability of the simulated VPR underlined by the large interdecile range, (ii) a strong difference between the reflectivity at the freezing level and its value in the liquid phase which translates into the need to use a different Z-R relationship for the conversion of snowy reflectivities into rain rates (Z-S relationship), and (iii) a non-constant simulated reflectivity towards the ground below the bright band revealing the evaporation/enhancement processes of rainfall.

All these statements illustrate the potential benefits of the POVPR method compared to the operational VPR correction.

## 4.3  Cold front of the 3rd March 2017

The second case presented in this study focuses on the cold front that passed through southwestern France on March 3rd and 4th 2017. The freezing level rapidly dropped from an altitude of about 2,800 m to roughly 1,200 m (and even lower on the Pyrenees foothills).

The POVPR method was applied to the operational reflectivities from the C-band polarimetric radar located in Bordeaux. The western side of the radar domain is offshore and the continental terrain is mainly flat, except in the extreme south and south-east parts of the domain where the Pyrenees and the Black Mountain reach 3,400 and 1,200 m above sea level, respectively (Fig. 4b). The radar coverage is almost perfect, with few beam blockages.

Figure 10 displays a snapshot of the situation through the radar reflectivity at the lowest elevation (0.4°) and the simulated

freezing level altitude from AROME-NWC, both at 21 UTC. The extent of the precipitating area and the relatively slow motion of the front (35 to 40 km h$^{-1}$) permitted significant accumulations (up to 40 mm) over the major part of the domain. The first part of the episode (18 to 00 UTC) is of particular interest as the operational radar QPE Panthere produced notably biased estimations (see Fig. 11a) in two distinct areas: (i) over the Pyrenees foothills and the adjacent plains (extreme south of the domain) where radar rainfall estimations are much lower than the rain-gauge accumulations, and (ii) over the western foothills

of the Massif Central range (far eastern part of the domain) where the radar largely overestimated the precipitation amounts.

A west-to-east vertical cross-section of the simulated reflectivity by AROME-NWC at 21 UTC (see Fig. 12a) shows that virga (precipitation evaporating before reaching the ground) are present ahead of the main precipitation core associated with the cold front. Once you move away from the radar, the altitude of the lower beam (0.4°) increases dramatically and cannot consequently sample the lower part of the atmosphere. In this case, virga are seen by the radar similarly as precipitation reaching





the ground level. Because the beam is above the freezing level at this range, the operational VPR correction reinforces the overestimation. In the example displayed in Figure 12, the extrapolation of the simulated reflectivity of the lowest elevation through the operational VPR correction would lead to a ground reflectivity of about 27 dBZ, that is to say a rain rate of 1.8 mm h$^{-1}$ with the Marshall-Palmer relationship used in the operational system (Z=200R$^{1.6}$). By using simulated VPRs in the vicinity

of each radar pixel, the POVPR method is able to take into account the evaporation of the precipitation during their falling in this area. As a result, the radar QPEs computed over the western foothills of the Massif Central range front are in much better agreement with the low accumulations measured by the rain gauges (see Fig. 11). Moreover, the new method is able to capture the enhanced precipitations over the Pyrenees foothills and adjacent plains. Note that the method has not been evaluated over the Pyrenees themselves because of the low freezing level altitude responsible for snowy precipitation and consequently lead-

ing to difficulties to evaluate rainfall amounts with non-heated rain gauges.

The improved performances of the POVPR method are confirmed by the scatter plots comparing hourly and 6-h accumulations with those measured by the rain gauges (Fig. 13). The RMSEs are reduced by 14 and 24% while the mean biases are divided by almost 2. The Pearson correlation coefficients jump from 0.63 for the hourly accumulations and 0.70 for the 6-h accumulations to 0.73 and 0.84, respectively. The Panthere QPE overestimations observed for low rain-gauge accumulations correspond to

the virga areas and are significantly mitigated with the POVPR method. The benefits of the method at greater range (beyond 100 km) are clearly illustrated by the Figure 8b. They are less evident at shorter distances. About 70% of mean hourly RMSEs at rain gauge locations are reduced compared to the Panthere QPE ones.

Similarly to the previous case, the median simulated and operational VPRs for the POVPR QPE and the Panthere QPE respectively at 2300 UTC (see Fig. 9b) differ significantly, in terms of shape but also intensity. The variability of the simulated VPRs

emphasizes the diversity of precipitation profiles over the whole radar domain, ranging from profiles with strong evaporation ahead of the cold front to very humid profiles in the main precipitating area.

## 5   Conclusions

Extrapolating rainfall at the surface from radar reflectivities measured at the beam altitude is very challenging. In operational weather services such as Météo-France, this is most commonly made thanks to a VPR correction uniformly applied over the

whole radar domain. The success of this method can be explained by the fact that it is almost fully observation-based (only the simulated freezing level altitude is generally used) and it is computationally efficient, with on average good improvements in the radar QPE. However, when the spatial and vertical variability of the precipitation are large, selecting a unique conceptual VPR becomes very inefficient.

The purpose of this study was to illustrate the potential benefits of a new approach that takes advantage of the simulated VPRs

from the NWP model AROME-NWC to perform a pixel-wise evaluation of the most probable rain rate at the ground from the radar reflectivities aloft. To our knowledge, this is the first time that such a method that combines model outputs and radar observations is used to derive QPEs. The implementation of this method on two stratiform situations (March 3$^{rd}$ 2017 and April 30$^{th}$ 2018 cases) yielded positive results compared to the current operational system.



In both situations the dramatic biases induced in the operational VPR correction, by either the overestimation of the freezing level altitude, or the lack of evaporation or precipitation enhancement below, or both, are largely mitigated in the new method. In both cases, the gains are significant up to maximum range, despite the high altitude of the radar beam. This is very encouraging for the application of the POVPR method in mountainous areas where the radars are most often installed far from the mountains

5   or at high altitude to limit the beam blockages. In addition to this, the use of high-resolution NWP models such as AROME-NWC promises to be very helpful for taking into account the high variability of the precipitation that is generally expected over complex terrains. It is indeed reasonable to expect that a NWP model will produce VPR profiles that take into account (i) the orientation of the slopes (windward or leeward) leading to an enhancement or a reduction of the precipitation at the ground, (ii) the strong spatial variability of low level humidity driving the evaporation process, as well as (iii) the higher wind direction

10   and speed variability that causes horizontal displacement of the precipitation as it falls.

This potential for improving QPE in mountainous regions will be evaluated in future work. The robustness of the method will also be tested over longer periods as well as the use of multiple model runs.

*Acknowledgements.* The authors wish to thank all members of the Météo-France Radar Centre (CMR) for their help all along this study, and particularly C. Augros for her assistance with the use of the radar forward operator and, J. Millet and M. Martet for their technical support.





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





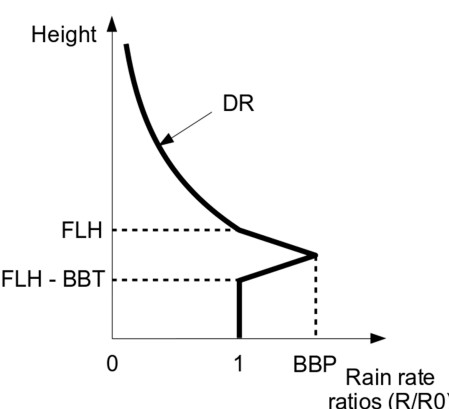

**Figure 1.** Idealized VPR used for the operational correction of reflectivities (Tabary, 2007) expressed in terms of rain rate ratios. FLH: freezing level height; BBP: bright band peak; BBT: bright band thickness; DR: decreasing rate above the freezing level.



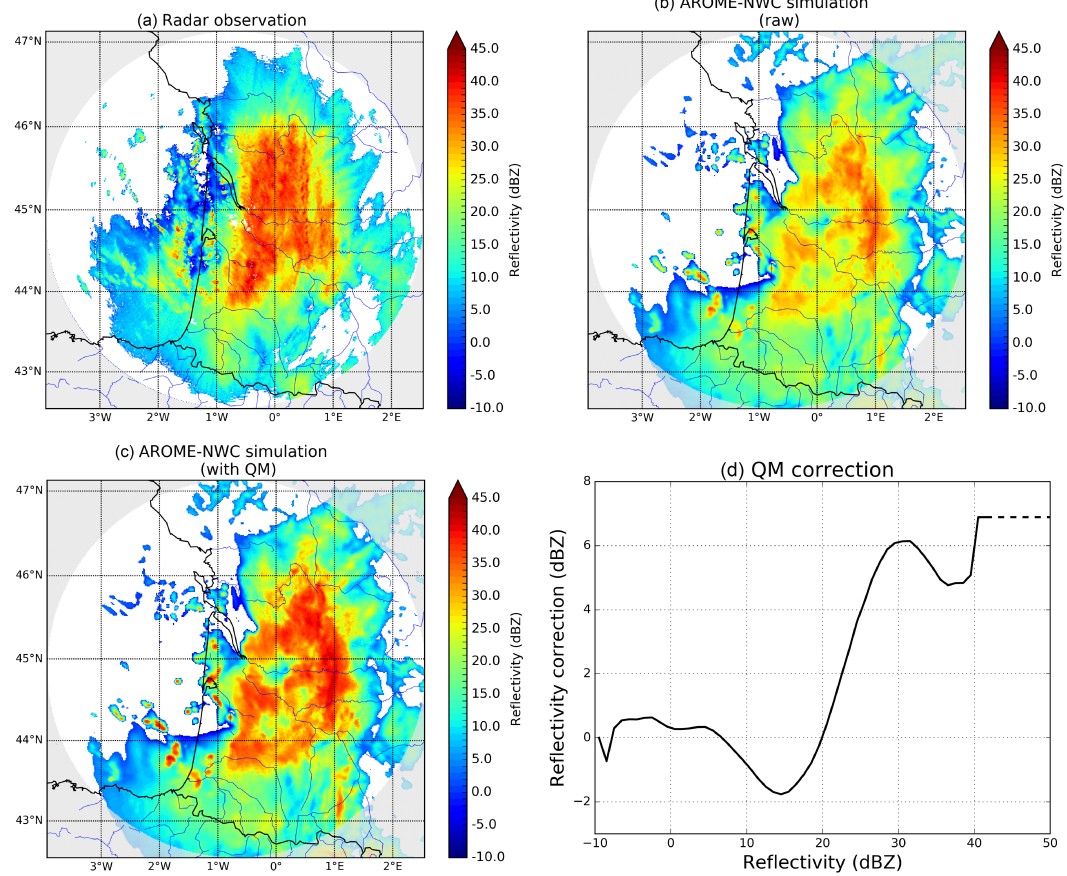

**Figure 2.** (a) Observed radar reflectivity from the radar of Bordeaux at the elevation 0,4° on March 3$^{rd}$ 2017 at 23 UTC. (b) Corresponding raw simulated radar reflectivity from AROME-NWC (run of 21 UTC). (c) Same as (b) but corrected by quantile mapping. (d) Reflectivity correction Cz resulting from the quantile mapping. The extension of the correction for higher reflectivities is represented in dashed lines.



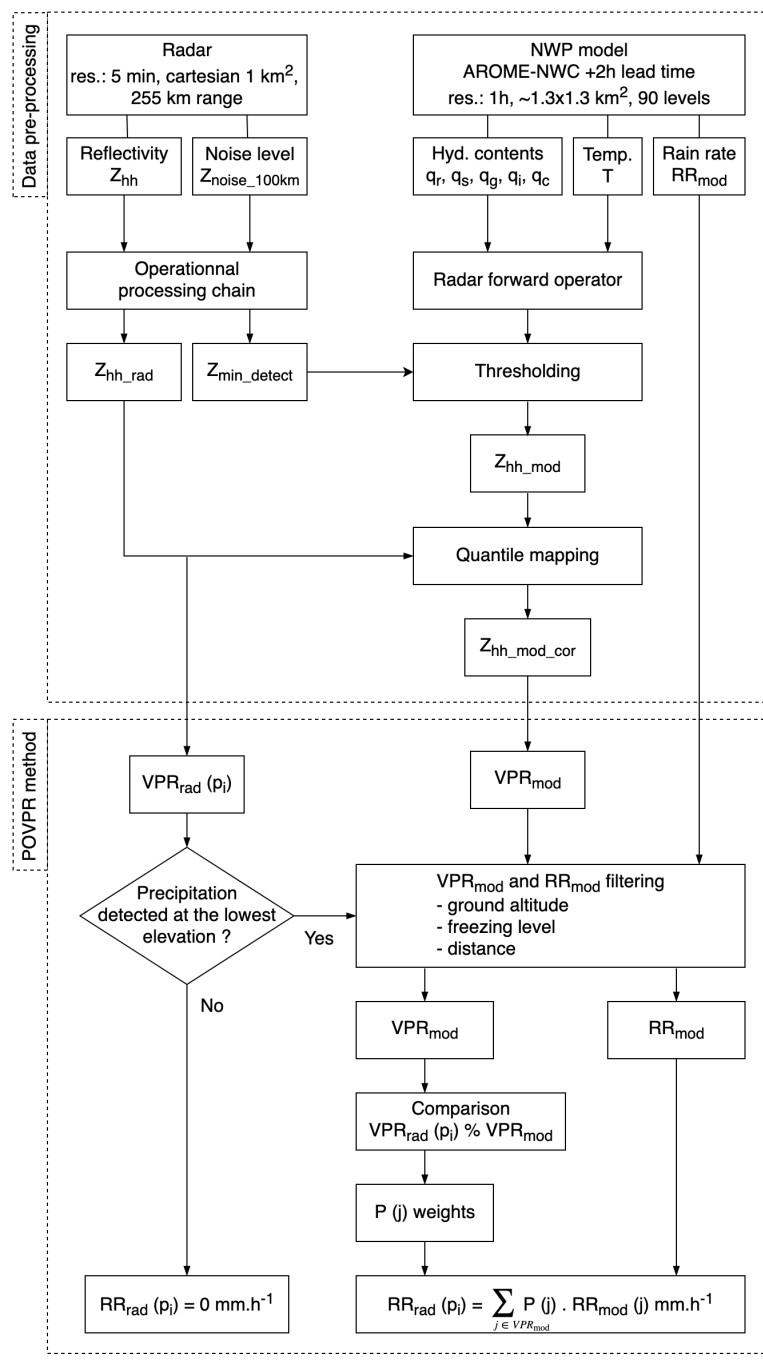

**Figure 3.** Process flow chart of the data pre-processing and the POVPR method for computing a pixel wise ground rain rate from observed and simulated reflectivities.



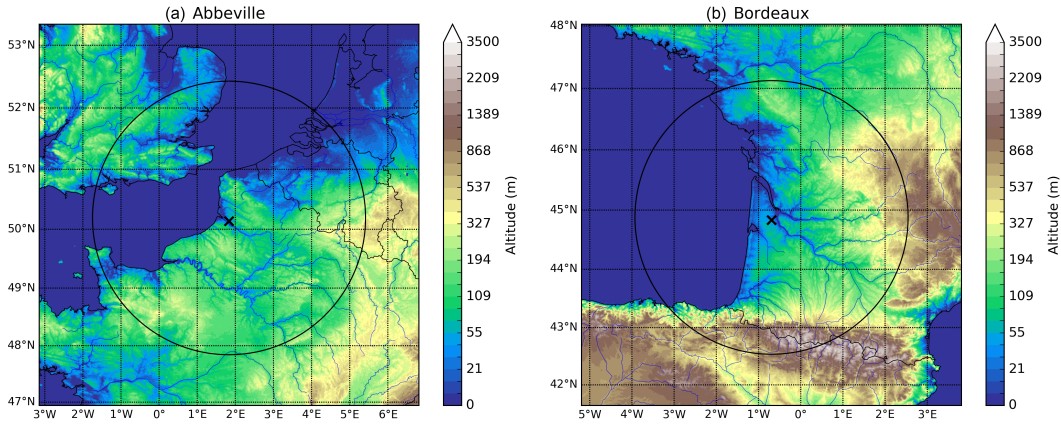

**Figure 4.** Topography in the Abbeville (a) and Bordeaux (b) radar areas. The radar positions are indicated by the black crosses and their 255 km ranges by the black circles. The domain displayed is extended by 100 km to represent the NWP model domain used for the computation of the POVPR accumulations.





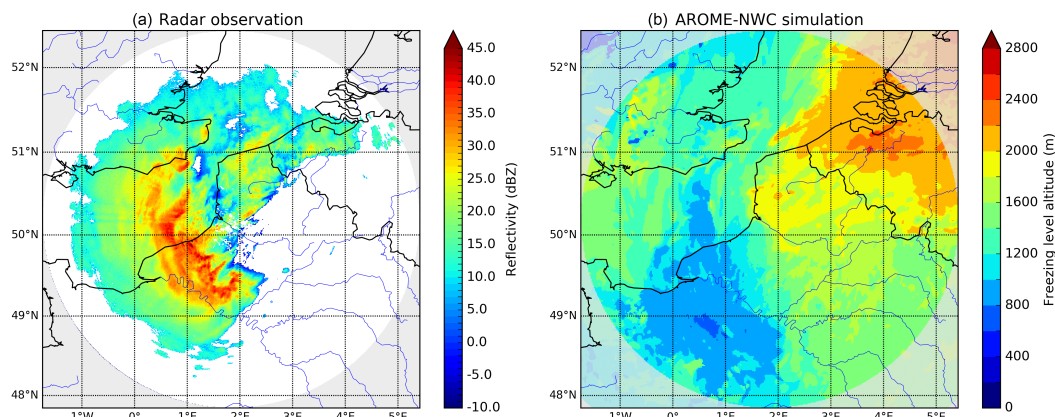

**Figure 5.** Observed reflectivity from the radar of Abbeville at the elevation 0.4° (a) and simulated freezing level altitude from the model AROME-NWC (+2 h lead time) (b) on April 30$^{th}$ 2018 at 06 UTC.



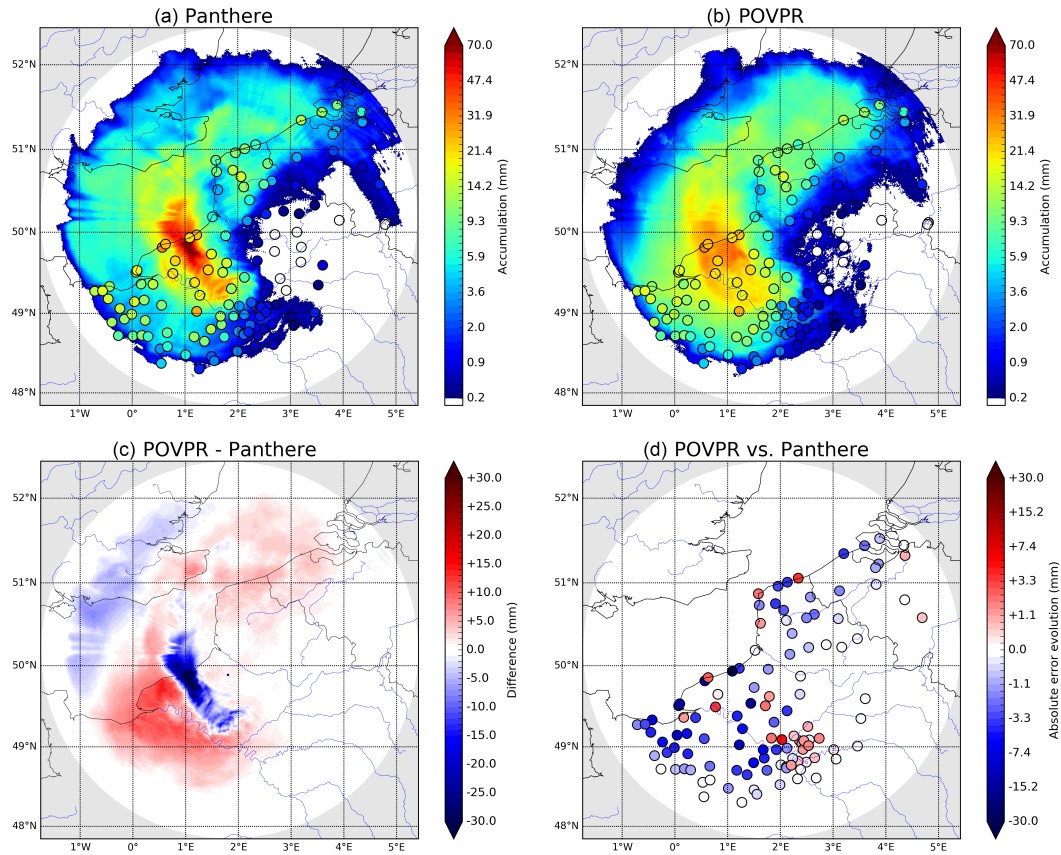

**Figure 6.** 6-h accumulations for the April 30$^{th}$ 2018 event between 03 and 09 UTC on the Abbeville radar domain (255 km range) with the operational method Panthere (a) and the POVPR method (b). Rain gauges accumulations where any precipitation has been detected by the radar are represented by circles. (c) Difference between both spatial accumulations (POVPR – Panthere). (d) Absolute error evolution between both methods at the rain gauge locations.




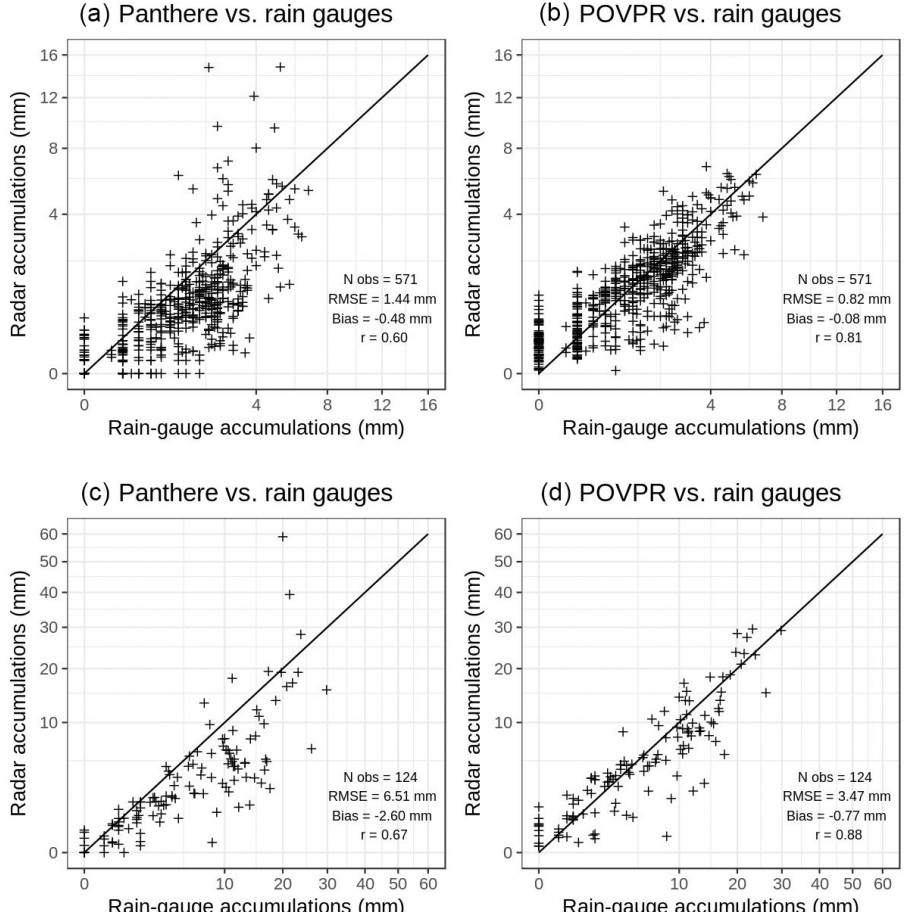

**Figure 7.** Comparisons between rain-gauge and radar accumulations computed for the April 30th 2018 event between 03 and 09UTC on the Abbeville radar domain. (a - b) hourly accumulations; (c - d) 6-h accumulations; (a - c) operational method Panthere; (b - d) POVPR method. Are also displayed the number of observations (N obs), the root mean square error (RMSE), the mean bias (Bias) and the Pearson correlation coefficient (r). Data where the lowest valid radar elevation did not detect any precipitation (no precipitation or beam too high) were removed from the data set.





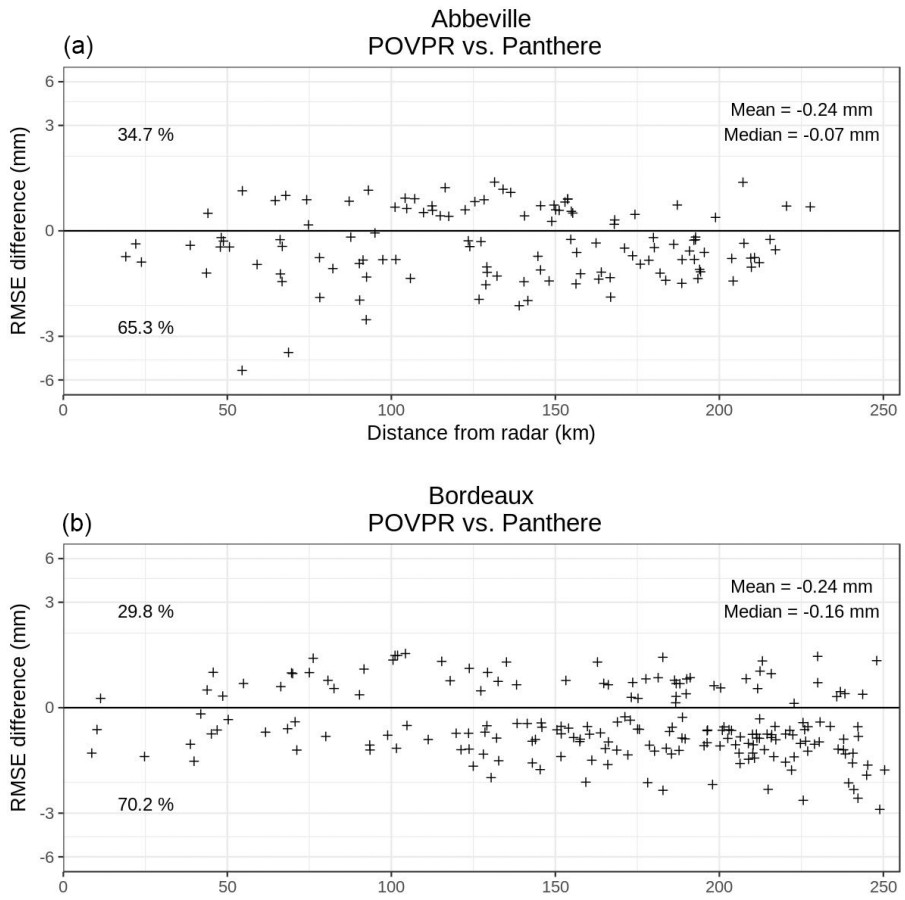

**Figure 8.** Differences between mean hourly RMSEs at the rain gauges locations from the POVPR QPE and from the Panthere QPE according to the distance from the radar, for the April 30[th] 2018 event between 03 and 09 UTC on the Abbeville radar domain (a) and for the March 3[rd] 2017 event between 18 and 00 UTC on the Bordeaux radar domain (b). Negative values indicate a better estimation of the POVPR QPE (lower RMSE), positive values means the opposite. The percentages of positive and negative values of RMSE difference are also indicated.





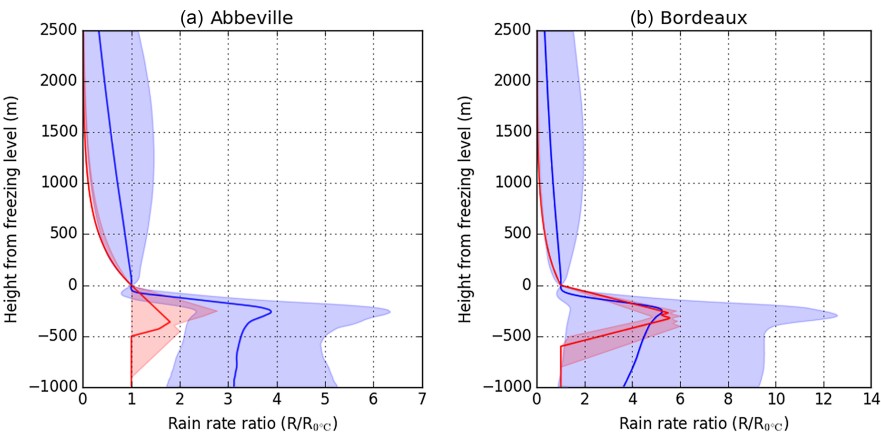

**Figure 9.** (a) Median VPR predicted by AROME-NWC for April 30[th] 2018 at 06 UTC on the Abbeville radar domain (blue) and median VPR of the VPRs used operationally between 0530 and 0630 UTC for the correction of reflectivities (red). VPRs are expressed as rain rate ratios, rain rate at the freezing level (R0°C) being the reference. Only profiles with a rain rate greater than 0.1mm h[-1] at the freezing level have been kept. The first and last deciles are delimited by the shaded areas. (b) Same as (a) for March 3[rd] 2018 at 23 UTC on the Bordeaux radar domain.



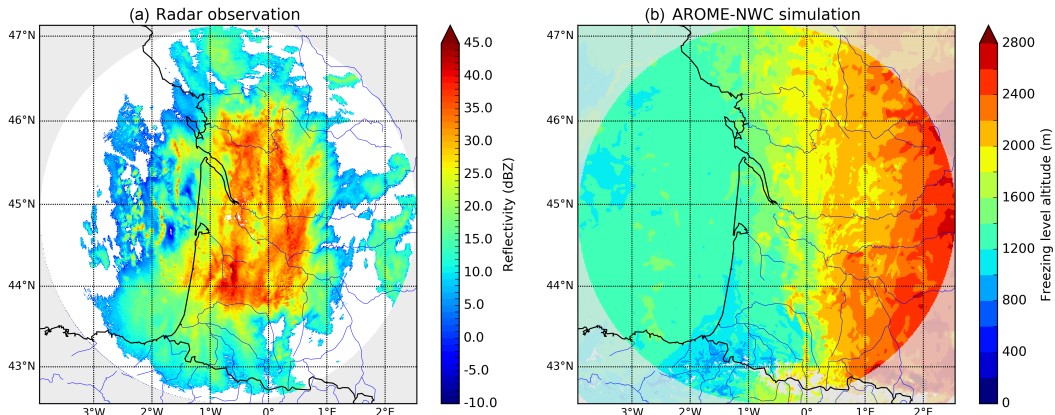

**Figure 10.** Same as Fig. 5 for March 3rd 2017 at 21 UTC on the radar of Bordeaux.

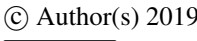



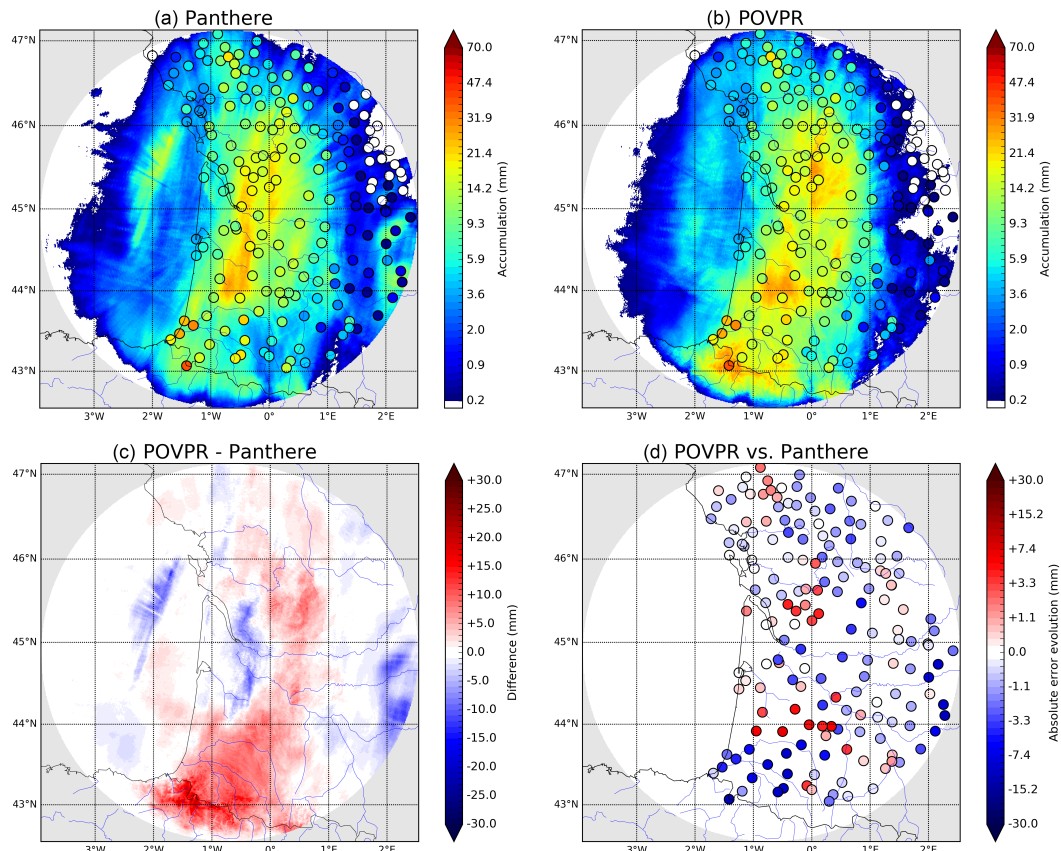

**Figure 11.** Same as Fig. 6 for the March 3$^{rd}$ 2017 event between 18 and 00 UTC on the Bordeaux radar domain.



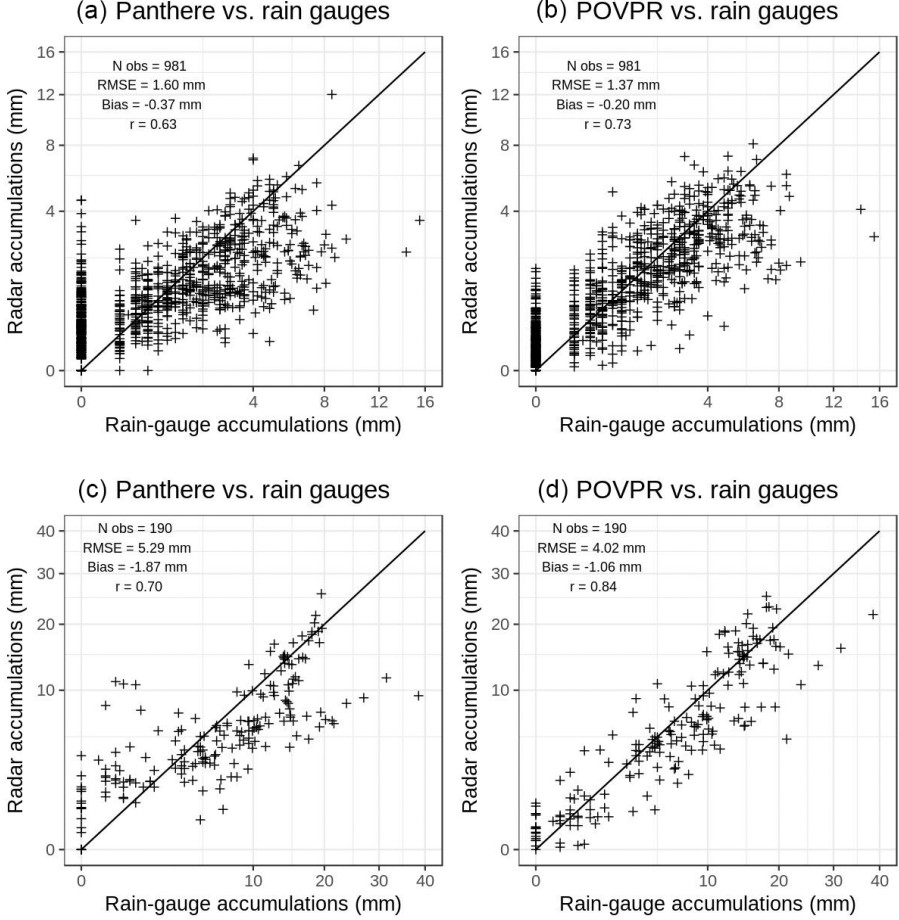

**Figure 12.** Same as Fig. 7 for the March 3$^{rd}$ 2017 event between 18 and 00 UTC on the Bordeaux radar domain.





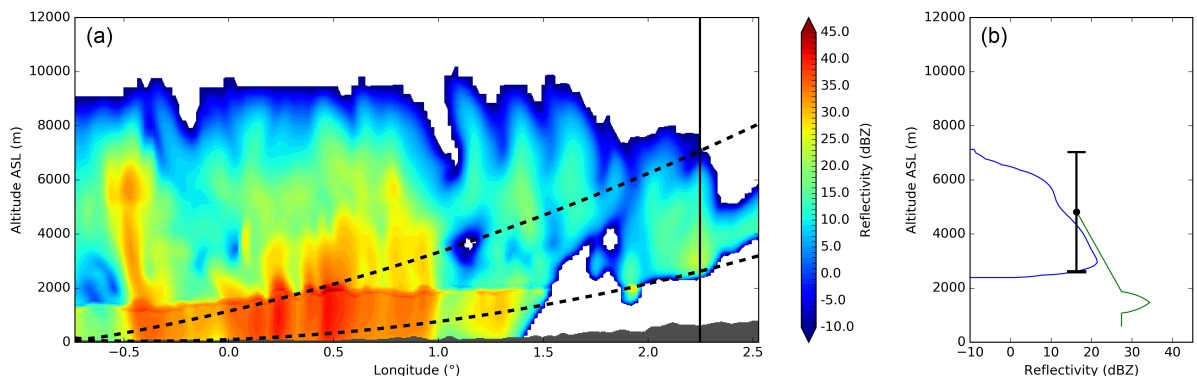

**Figure 13.** (a) West-to-east cross-section of the simulated reflectivity from AROME-NWC (after quantile mapping) of the 3$^{rd}$ March 2017 at 21 UTC passing by the radar location (situated on the bottom left corner). The radar beam aperture (1.1°) of the lowest elevation (0.4°) is represented by the black dashed lines. (b) Simulated VPR (blue) and extrapolated simulated VPR computed with the operational method from the lowest radar elevation (0.4°) (green) at the location indicated by the vertical black line on the cross-section (a). The radar beam aperture (1.1°) of the lowest elevation (0.4°) is represented by the black segment.