# Peer review of "Combined use of volume radar observations and high-resolution numerical weather predictions to estimate precipitation at the ground: methodology and proof of concept"

_Atmospheric Measurement Techniques, 2019_

## Referee Comment (RC1) · Anonymous Referee #1 · 18 Jun 2019

This paper presents an initial proof of concept of a new way to tackle the issue of the retrieval of the vertical profile of reflectivity (VPR), an important step in the radar estimation of precipitation on the ground. The method essentially consists in using the VPR estimated from the model which is most similar to the VPR observed from the radar at any given radar range gate. Currently, most operational VPR estimation methods provide a single VPR that is applied to the whole radar domain. The presented method outperforms the status quo mainly by providing an spatialized VPR and by being able to provide information of the VPR at low altitudes that are not observable by the

[Figure]

**[AMTD](AMTD)**
radar. Those characteristics ensure that the new method provides better performance particularly in situations where there is a large variability of the melting layer height within the radar domain or there is poor visibility of the radar at low altitudes. The paper covers and important topic, has significant novelty and it is well written. Therefore I recommend its publication provided some minor issues are addressed.

General comments:

1. As it is presented the method is rather complex and computationally expensive. The authors should provide an estimation of the computational cost and discuss its implications on its operational implementation.

2. The method is highly dependent on the performance of the NWP. However I can imagine situations where the forecast of quantitative precipitation by the model performs poorly, particularly in convective situations where the model may have difficulties in forecasting the right air temperature and generating convective cells. I understand that this paper is basically a proof of concept and that more analysis has to be carried out but I would appreciate a discussion of the possible limitations of the presented methodology and some suggestions on how to tackle them. For example, what to do when no VPR from the model is similar enough to the radar estimated VPR? What are the consequences of having a very poorly sampled observed VPR? That is a situation that may happen regularly in areas where the radar has poor visibility such as in the mountains.

Specific comments:

Page 2 Line 3 – I would change backscattered power by backscattered signal since the phase also plays and important role in modern radars.

Page 2 Line 24 - . . . NWP models . . .

Page 3 Line 16 - . . . between 0 and 3°)

Page 3 Line 17 - raw elevation scans . . .

Page 3 Line 19 – (Figueras i Ventura and Tabary, 2013)

Page 3 Line 20 – . . . on a regular Cartesian grid . . .

Page 6 Line 20 – Assuming that the iso-0°C isotherm of the model at the radar range gate location is essentially correct is a strong constraint of the method since the position of the air mass can be shifted by several kilometres or the temperature may not be forecasted correctly. This should be highlighted and discussed in more detail.

Page 7 Line 28 - . . . For the purpose of this study, . . .

Page 13 Line 36 – The name of the first author is repeated twice (Georgiou S.)

Fig. 12 and Fig. 13 – The reference to those two figures in the text have been swapped.

---

## Referee Comment (RC2) · Anonymous Referee #2 · 17 Jul 2019

The paper present a proof of concept on the use of NWP forecast to support and enhance radar QPE through an innovative way to define the vertical profile of reflectivity (VPR) to correct the estimation of surface reflectivity from upper layer observed one.

Further the method is able to provide a spatial variability for the VPR overcoming one of the limit of the standard techniques use to define VPR.

Major limitation of this method rely on NWP performances. Only two events are discussed in the paper and seems that NWP perform quite well in both, whats happen if

[Figure]

NWP does not perform well? Could be important to test the method over a wide range of meteorological events in order to understand and compare this method with Panthere QPE. Nevertheless, as clearly stated in the paper title, this is a proof of concept so we can refer and discuss only the idea, leaving a full demonstration and comparison to a next step.

For this reason I recommend to publish this paper with some minor issues.

Specific comments:

1) Page 31 line 21 Please clarify better what means "All the scans .... advection field." Are radar scans not synchronous?

2) page 3 line 23 It is stated that 300 profiles are obtained varying 4 parameters. Could you explain how do you change the parameters .? ( ie random choice within a predefined range for each parameter)

3) page 3 line 24 "compared with observed rain rate accumulation ratio" Do you compare with radar rain rate or with others data? please specify.

4) page 4 lines 6-7 The sentence "Thus, to compensate ....QPE product." is a speculation could be not true the the difference comes only from the VPR limitation. Please reformulate the sentence.

5) page 5 line 3 Please provide a range of variability for "delta" (threshold to reject noisy pixel).

6) page 5 lines 8-10 The laste sentence of the paragraph seems to be a personal guess, not support by any data,. please reformulate.

7) page 6 - line 11 Sect 1.1.1 ????? Could be a wrong reference to the text.

8) page 7 line 2 Drad - You aggregate all 5-min radar reflectivity during the hour centered around the model time. This seems to be a limit in a quite fast changing freezing level height event. Could you comment please.

9) page 8 figure 3 This figure is very helpful to understand the method. I strongly suggest to move this figure to the begin of section 3.

10) page 10 line 11 As comment 7. Sect 1.1.3 ?????

11) page 9 -line 9 Add "observed" before "reflectivity". From this sentence I understand that in fig. 5 i will see "simulated Z" from AROME-NWC. From figure caption I read "observed Z". This is misleading.

12) page 9 line 19 I'm not sure that Sect 2.2 is correct or is as comment 7.

13) page 9 line 34 As comment 7 Sect 1.1 ????

14) page 10 line 33 —- page 11 line 4 I strongly suggest to reformulate this text. It is much more close to a newspaper article than a scientific one. There is nothing dramatic (refer line 33) in the altitude reached by the radar beam. This is a propagation effect well known. Further overshooting as well as evaporation below the sampling height are clearly highlighted in any radar school book.

15) page 10 line 31 - Figure 12 and figure 13 In the text is referred figure 12a, but it refer to fig 13a. Same for any time where figure 12 and fig 13 are used. Please correct.

---

## Author Comment (AC1) · 14 Aug 2019

The reply report to referees is attached in pdf document and also contains the new revised manuscript with changes tracking.

Please also note the supplement to this comment: https://www.atmos-meas-tech-discuss.net/amt-2019-166/amt-2019-166-AC1-supplement.pdf

---

## Author Comment (AC2) · 14 Aug 2019

**AMT-2019-166:**
**Responses to reviewers**

T. Le Bastard, O. Caumont, N. Gaussiat and F. Karbou

August 13, 2019

Dear Editor,

Please find below the answers to the referees comments on the manuscript number AMT-2019-166 and entitled "Combined use of volume radar observations and high-resolution numerical weather predictions to estimate precipitation at the ground: methodology and proof of concept".

We wish to thank the referees for their helpful remarks and suggestions. We have taken into account all comments in the revised manuscript. We hope that the revised manuscript now meets the journal requirements.

Best regards

Tony Le Bastard and co-authors

**Comments by Anonymous Reviewer #1**

> *1 - As it is presented the method is rather complex and computationally expensive. The authors should provide an estimation of the computational cost and discuss its implications on its operational implementation.*

The focus of this paper is to demonstrate the relevance of combining radar volume scans and NWP simulations of reflectivity to derive rain accumulations at the ground level. Our initial goal was to verify the feasibility of the study. We agree that the operational implementation aspect is very important but we did not deal with it explicitly in this paper. This work is fully prospective and the requirements for a potential operational implementation will be discussed in an upcoming paper.

Nevertheless we can already give some indications about this aspect (costs, the different improvements needed...):

The first step of the algorithm, which consists in producing simulated polarimetric variables from AROME-NWC outputs, runs in about 30 min for a 1h time step over a domain extending up to 355 km around the radar and at the full horizontal resolution ($1.3 \times 1.3$ km$^2$). Operationally, the AROME-NWC outputs are available about 30 min after analysis time. As we use +2h forecasts every hour, the computing time of the forward operator is still compatible with operational constraints. However, the performances could be further improved by parallelizing the code, segmenting the domain, down-sampling the raw model data.

The optimization of the algorithm searching for the most probable simulated profile and computing the corresponding rain rate accumulations is much more challenging. Indeed, each pixel of the radar field needs to be treated separately and faced with up to 1 200 simulated profiles. For now, it takes about 12h to compute a 1h accumulation over the entire radar domain (255 km range). This computation time is obviously not suitable with an operational implementation. Again, parallelizing the code could improved the performances. A more complex analysis of the simulated dataset could also be performed previously to select the most appropriate profiles and avoid redundancies. This could be done for instance by clustering the similar profiles. Finally, the use of 3-dimensional reflectivity field projected on a regular grid could facilitate the comparison with simulated data.

The following text has been added at the end of the Section 3.1:

"Reducing the number of candidates helped to run the algorithm in a reasonable time for research (about 12h for a 1h accumulation over the entire radar domain). The performances of our algorithm could be highly improved by parallelizing the processing (for the moment each radar pixel is processed separatly) to meet operational requirements. The use of clustering technics (to select the most suitable profiles) as well as the use of 3-dimensional reflectivity field projected

on a regular grid could also considerably improve the algorithm performances."

> *2 - The method is highly dependent on the performance of the NWP. However I can imagine situations where the forecast of quantitative precipitation by the model performs poorly, particularly in convective situations where the model may have difficulties in forecasting the right air temperature and generating convective cells. I understand that this paper is basically a proof of concept and that more analysis has to be carried out but I would appreciate a discussion of the possible limitations of the presented methodology and some suggestions on how to tackle them. For example, what to do when no VPR from the model is similar enough to the radar estimated VPR? What are the consequences of having a very poorly sampled observed VPR? That is a situation that may happen regularly in areas where the radar has poor visibility such as in the mountains.*

Indeed, this is a pertinent remark. But, the large extension of the domain (100 km) for the search of the most resembling simulated VPRs on one side, and the use of forecasts pretty close to the analysis time (+ 2h) in which radar data have been assimilated, largely mitigates the risk of using unrealistic simulation data. However, a situation where the model performs poorly can still happen. To tackle this issue, we would suggest to use the current operational QPE (Panthere) as a fallback solution.

Here is a complement introduced at the end of the Section 3.3:

"To deal with the possible poor performances of the NWP model, we could use the current operational algorithm as a fallback solution. For each observed radar apparent VPR, the simulated profiles are linearly combined according to the cost function described in eq. (3) and (4). In a situation with a low level of similarity between observed and simulated profiles, the $P$ weights used for the linear combination are rather low. As a solution, we could introduce an additional term depending on the rain rate produced by the current operational algorithm (Panthere) in the linear combination of the simulated rain rates (eq. 10). The features of this new term would be negligible when many resembling simulated profiles have been found and become predominant when the $P$ weights fall down. More simply, we could use the Panthere QPE when the sum of $P$ weights falls below of a threshold to be defined. With such a fallback procedure, in case of a poor simulation, the retrieved accumulation would be at least relatively close to the current operational one."

> *3 - Page 2 Line 3 - I would change backscattered power by backscattered signal since the phase also plays and important role in modern radars.*

The reviewer is right. The sentence has been modified:

"... which is the backscattered signal  of hydrometeors in the atmosphere."

> *4 - Page 2 Line 24 - ... NWP models ...*

Corrected: "NWP models"

> *5 - Page 3 Line 16 - ... between 0 and 3°) ...*

Corrected: "between 0  and 3°)"

> *6 - Page 3 Line 17 - ... raw elevation scans ...*

Corrected: "raw elevation scans"

> *7 - Page 3 Line 19 - ... (Figueras i Ventura and Tabary, 2013) ...*

Corrected. ¿¿¿ "(Figueras i Ventura and Tabary, 2013)"

> *8 - Page 3 Line 20 - ... on a regular Cartesian grid ...*

Corrected: "on a regular Cartesian  grid"

> *9 - Page 6 Line 20 - Assuming that the iso-0°C isotherm of the model at the radar range gate location is essentially correct is a strong constraint of the method since the position of the air mass can be shifted by several kilometers or the temperature may not be forecasted correctly. This should be highlighted and discussed in more detail.*

This is absolutely true. However, the mis-representation of 0°C isotherm is already a significant problem in the operational method since it only uses a unique VPR, conditioned by a unique 0°C isotherm estimated from NWP global model (ARPEGE) outputs in the vicinity of the radar and refined with the cross-correlation coefficient $\rho_{hv}$ (Tabary et al., 2006). In the new method we use the AROME analysis which is performed every hour and that is therefore more likely to represent the true spatial and temporal variability of the 0°C isotherm.

The following sentence (in blue) has been added:

"The 0°C isotherm at $p_i$ location is estimated by the one of the co-located

point of the AROME analysis. Model displacement errors could introduce errors in the determination of the correct 0°C isotherm, but, the AROME analysis is performed every hour and can represent the freezing with a smaller error than the one induced by the use of a unique value for the entire domain in the current operational method."

> *10 - Page 7 Line 28 - ... For the purpose of this study, ...*

Corrected: "For the purpose of this study,"

> *11 - Page 13 Line 36 - The name of the first author is repeated twice (Georgiou S.)*

Corrected: "Georgiou, S., Gaussiat, N., and Lewis, H., and Georgiou, S.:"

> *12 - Fig. 12 and Fig. 13 - The references to those two figures in the text have been swapped.*

Corrected. Indeed, the references to Figure 12 and 13 had been swapped.

**Comments by Anonymous Reviewer #2**

> *1 - Page 31 Line 21 - Please clarify better what means "All the scans ....*
> *advection field." Are radar scans not synchronous?*

The radar volume scans are produced successively during a cycle lasting 5 minutes. Once proceeded, all the scans are synchronized temporally to the end time of the cycle by using an advection field.

> *2 - page 3 line 23 It is stated that 300 profiles are obtained varying 4 parameters. Could you explain how do you change the parameters .? ( ie random choice within a predefined range for each parameter)*

To improve clarity, the text has been completed as follows:

"By varying the four parameters within predefined ranges ($\pm$ 200 m for FLH, 1 to 6 for BBP, 200 to 800 m for BBT and -1.5 to -6 dB km$^{-1}$ for DR), 288 ratio profiles of rain rate (deduced from the Marshall-Palmer relationship $Z = 200R^{1.6}$) are built and compared to the observed radar rain rate accumulation ratios."

> *3 - Page 3 Line 24 - "compared with observed rain rate accumulation ratio". Do you compare with radar rain rate or with others data? Please specify.*

See response to item 2.

> *4 - Page 4 Lines 6-7 - The sentence "Thus, to compensate ....QPE product." is a speculation could be not true the the difference comes only from the VPR limitation. Please reformulate the sentence.*

The following reformulation has been introduced to the revised manuscript:

"These limitations are partially overcome in the operational processing of the final 5-min QPE product by an adjustment using hourly rain gauge and radar data from the past hours (up to 40 h)."

> *5 - Page 5 Line 3 - Please provide a range of variability for "delta" (threshold to reject noisy pixel).*

The following sentence has been added:

"$\delta$ typically ranges between 1 and 2 dB."

*6 - Page 5 lines 8-10 - The last sentence of the paragraph seems to be a personal guess, not support by any data,. please reformulate.*

This proposition is a reformulation of some conclusions from Augros et al. (2016). As they mentioned, the different resolution of radar ($1 \times 1$ km$^2$) and model data ($2.5 \times 2.5$ km$^2$) could explain the differences found in the lower levels. However, as far as we know, no further investigations on this particular fact have been done yet.

*7 - Page 6 - Line 11 - Sect 1.1.1 ????? Could be a wrong reference to the text.*

Corrected: "(see eq. 1 )"

*8 - Page 7 Line 2 - $D_{rad}$ - You aggregate all 5-min radar reflectivity during the hour centered around the model time. This seems to be a limit in a quite fast changing freezing level height event. Could you comment please.*

Since the data aggregated in $D_{rad}$ come from all radar volume scans at different range, altitude and time, the influence of a time- and space-varying freezing level height is mitigated. However, it could be non-negligible in a case where, for example, $D_{rad}$ is made of pure snowy reflectivities whereas $D_{mod}$ also contains data from the liquid phase. Then, the $D_{mod}$ distribution would probably extend to higher values than the $D_{rad}$ one and it would impact the bias correction. Ideally, the quantile mapping correction should be made, not only according to the reflectivity, but also according to the temperature thus taking into account different phases of hydrometeors, as well as the distance for considering the beam broadening. The use of multiple runs for $D_{mod}$ could also help to mitigate the potential bad estimation of the freezing level height.

The following sentence has been added to the revised version:

"Ideally, the quantile mapping correction should be made, not only according to the reflectivity, but also according to the temperature for taking into account the different phases of hydrometeors and their different response in terms of reflectivity, as well as the distance for considering the beam broadening."

*9 - Page 8 figure 3 - This figure is very helpful to understand the method. I strongly suggest to move this figure to the begin of section 3.*

You are totally right. A reference has been added at the beginning of Section 3:

For an easy following of the different steps of the method described in this section, the reader can refer to the Figure 3.

> *10 - Page 8 Line 17 - As comment 7. Sect 1.1.3 ?????*

Corrected: "(see Sect 2.1 )"

> *11 - Page 9 -Line 9 - Add "observed" before "reflectivity". From this sentence I understand that in fig. 5 i will see "simulated Z" from AROME-NWC. From figure caption I read "observed Z". This is misleading.*

Corrected: "Figure 5 displays the observed reflectivity"

> *12 - Page 9 Line 19 - I'm not sure that Sect 2.2 is correct or is as comment 7.)*

Corrected: "(see Sect 3.1 )"

> *13 - Page 9 Line 34 - As comment 7 Sect 1.1 ????*

Corrected: "(see Sect 3.1 )"

> *14 - Page 10 Line 33 - Page 11 Line 4 - I strongly suggest to reformulate this text. It is much more close to a newspaper article than a scientific one. There is nothing dramatic (refer line 33) in the altitude reached by the radar beam. This is a propagation effect well known. Further overshooting as well as evaporation below the sampling height are clearly highlighted in any radar school book.*

You are right. In saying "dramatically", we thought "significantly". Obviously this phenomenon and the consequences are pretty well known. To avoid confusions, "dramatically" has been replaced by "significantly":

"Once you move away from the radar, the altitude of the lower beam $(0.4°)$ increases significantly  and cannot consequently sample the lower part of the atmosphere."

> *15 - Page 10 Line 31 - Figure 12 and figure 13 In the text is referred figure 12a, but it refer to fig 13a. Same for any time where figure 12 and fig 13 are used. Please correct.*

Corrected. Indeed, the references to Figure 12 and 13 had been swapped.

[revised manuscript text omitted]

Reducing the number of candidates helped to run the algorithm in a reasonable time for research (about 12h for a 1h accumulation over the entire radar domain). The performances of our algorithm could be highly improved by parallelizing the processing (for the moment each radar pixel is processed separately) to meet operational requirements. The use of clustering technics (to select the most suitable profiles) as well as the use of 3-dimensional reflectivity field projected on a regular grid could also considerably improve the algorithm performances.

**3.2 Model bias correction**

To maximise the chances of finding the simulated apparent VPRs that best fit the observations, a model bias correction is used to bring a maximum number of simulated observations as close as possible to the observed ones. Simulated observations can be biased either because the model itself is biased (approximation in the model physics, representativeness errors) or because the radar forward operator is biased or both. For simplicity, all the model biases are corrected by applying a quantile mapping correction (QM), a method commonly used in climatic simulations (Lafon et al., 2013). Thus, this correction is applied every hour and is used to match the distributions ($D_{mod}$) of the simulated reflectivities produced by the model ($Z_{hh\_mod}$) with the observed distributions ($D_{rad}$) computed by aggregating all 5-min radar reflectivity scans during the hour centred around the model time. Ideally, the quantile mapping correction should be made, not only according to the reflectivity, but also according to the temperature for taking into account the different phases of hydrometeors and their different response in terms of reflectivity, as well as the distance for considering the beam broadening. The chosen 1h time window ensures that, the range of values of each 5-min observed reflectivity data set processed by the POVPR algorithm, is covered by the closest in time simulated reflectivity data set used for the VPR estimation. Some tests (not presented here) have shown that a longer temporal window gives poorer final QPE results.

[revised manuscript text omitted]

To deal with the possible poor performances of the NWP model, we could use the current operational algorithm as a fallback solution. For each observed radar apparent VPR, the simulated profiles are linearly combined according to the cost function described in eq. (3) and (4). In a situation with a low level of similarity between observed and simulated profiles, the $P$ weights used for the linear combination are rather low. As a solution, we could introduce an additional term depending on the rain rate produced by the current operational algorithm (Panthere) in the linear combination of the simulated rain rates (eq. 10). The features of this new term would be negligible when many resembling simulated profiles have been found and become predominant when the $P$ weights fall down. More simply, we could use the Panthere QPE when the sum of $P$ weights falls below of a threshold to be defined. With such a fallback procedure, in case of a poor simulation, the retrieved accumulation would be at least relatively close to the current operational one.

[revised manuscript text omitted]

**Figure 13.** (a) West-to-east cross-section of the simulated reflectivity from AROME-NWC (after quantile mapping) of the 3[rd] March 2017 at 21 UTC passing by the radar location (situated on the bottom left corner). The radar beam aperture (1.1°) of the lowest elevation (0.4°) is represented by the black dashed lines. (b) Simulated VPR (blue) and extrapolated simulated VPR computed with the operational method from the lowest radar elevation (0.4°) (green) at the location indicated by the vertical black line on the cross-section (a). The radar beam aperture (1.1°) of the lowest elevation (0.4°) is represented by the black segment.